# Medications for the Treatment of Alcohol Dependence—Current State of Knowledge and Future Perspectives from a Public Health Perspective

**DOI:** 10.3390/ijerph20031870

**Published:** 2023-01-19

**Authors:** Iga Stokłosa, Gniewko Więckiewicz, Maciej Stokłosa, Magdalena Piegza, Robert Pudlo, Piotr Gorczyca

**Affiliations:** Department and Clinic of Psychiatry, Medical University of Silesia, 42-612 Tarnowskie Góry, Poland

**Keywords:** alcohol dependence, medications, treatment

## Abstract

No single effective therapy for alcohol abuse has been found, despite it being a serious sociological and economic problem for hundreds of years. It seems difficult to find a single drug as a panacea for the alcohol problem due to the complexity of the pathophysiology of alcohol dependence. The purpose of this narrative review is to review existing and potentially future pharmaceuticals for the treatment of alcohol dependence in the most affordable way possible. Psychotherapy is the mainstay of treatment for alcoholism, while few drugs approved by legislators are available in the augmentation of this treatment, such as acamprosate, disulfiram, and naltrexone, approved by the FDA, and nalmefene by the EMA. There are recent reports in the literature on the possibility of using baclofen, topiramate, varenicline, and gabapentin in the treatment of alcohol dependence. Moreover, the results of recent clinical trials using psychoactive substances such as psilocybin and MDMA appear to be a breakthrough in the modern treatment of alcohol abuse. Despite this initial optimism, a lot of scientific effort is still needed before new pharmacological methods supporting the treatment of alcohol dependence syndrome will be widely available.

## 1. Introduction

According to data published by the WHO, three million deaths (5.3% of deaths) are caused annually by AUD (alcohol use disorder) [1]. In the United States, alcohol abuse affects approximately 6% of the population, 1 in 12 men and 1 in 25 women struggle with this health problem [2]. The diagnostic criteria for AUD include: a strong desire or sense of compulsion to consume alcohol, difficulty controlling substance use, withdrawal symptoms when substance use is reduced or discontinued, neglect of other alternative pleasures, and the continued substance use despite clear evidence of harmful consequences [3]. Alcohol abuse has many harmful consequences, economic, social, and health-related. According to a meta-analysis that reviewed 29 studies from around the world, the average economic cost of alcohol use worldwide is approximately 2.6% of the GDP (gross domestic product). One-third of the costs of alcohol consumption are direct costs, including health care, alcohol-related crime, and drunk driving accidents, while the other two-thirds are indirect costs, which include lost productivity due to absenteeism and premature mortality [4]. Although some studies have claimed that a low alcohol consumption of 0.1–7 drinks per week may have a protective effect on the cardiovascular system, these data come from epidemiological studies. However, it should be taken into account that alcohol consumption can lead to oxidative stress, apoptosis, mitochondrial dysfunction, and anatomical damage to the heart and circulatory system [5]. The effect of any alcohol and its episodic consumption on triggering gout attacks has been repeatedly demonstrated [6]. According to a report published by the WHO, alcohol abuse negatively affects cancer, heart disease, dementia, and the risk of suicide attempts, and also increases the number of car accidents [7]. Not only alcohol consumption, but also a hangover has an impact on the impairment of short-term memory as well as on the deterioration of sustained attention and psychomotor skills, as shown in a systematic review [8].

Despite the fact that alcohol has been around and consumed for centuries, there is still no single effective tool to combat addiction. The main tools are group therapies in Alcoholics Anonymous, which focus on the 12-step program and non-pharmacological interventions, while medications that can reduce the craving for alcohol and the addict’s desire for alcohol are still under-prescribed [9]. Among other non-pharmacological methods of influencing patients who are addicted to alcohol, peer-delivered recovery support service seems to be a relatively effective intervention worth further research [10]. The therapies that are currently implemented aim to promote total abstinence in addicts, but recently, controlled drinking, which is referred to as a harm-reduction method, seems to be a more attractive option [11]. To date, only three medications have been approved by the US Food and Drug Administration (FDA) for the treatment of alcohol dependence syndrome: acamprosate, disulfiram, and naltrexone. The European Medicines Agency (EMA) also approves nalmefene for treatment in addition to the three aforementioned agents [12]. Recently, there have been reports on the use of baclofen and the promising results of the introduction of topiramate, varenicline, and gabapentin into treatment [13]. Due to the heterogeneity of the alcohol dependence syndrome, it seems difficult to implement the concept that a single drug, used systematically and acting on a single mechanism in the body, could be effective in eliminating a long-standing alcohol problem in a patient [12]. Studies on the aspects related to alcohol consumption, such as alcohol-related organ damage, alcohol tolerance, and physical dependence on alcohol, have been conducted using animal models. The studies used, among others, operant conditioning, and attempts were made to identify genes influencing the predisposition to alcoholism. In addition, the effectiveness of drugs used in the treatment of AUD was also tested in animal models [14,15,16].

## 2. Materials and Methods

The aim of this narrative review is to highlight the existing, as well as new, future possibilities of pharmacotherapy, to support the treatment of alcohol dependence and to make the manuscript readable not only for physicians trained in reading scientific data, but for all those working in the field of public health. To this end, we searched PubMed for information on each substance, adding other terms such as “alcohol dependence”, “alcohol abuse”, or “alcoholism”. From these articles, we selected the most recent ones that were appropriate according to our scientific and clinical knowledge. Only articles in English were selected. In some parts, we added information from the internet, but only from reliable sources such as the website WHO or the official FDA prescribing information.

## 3. Results

We found four therapies (acamprosate, disulfiram, naltrexone, and nalmefene) registered for use in treating alcohol dependence, four therapies (baclofen, topiramate, varenicline, and gabapentin) not yet registered, and two (psylocybin and MDMA/Ecstasy) possible future therapies.

### 3.1. Acamprozate

Acamprosate is one of the drugs approved by the FDA and EMA for the treatment of alcohol dependence [13]. It is a well-tolerated and relatively safe drug that has been available for the treatment of alcohol dependence syndrome since 1989 [17]. The most common side effects of this drug include accidental injury, anxiety and depression, asthenia, pain, anorexia, stomach upset, dizziness, dry mouth, insomnia, itching, and sweating. It is a drug that can be safely used in individuals with hepatic insufficiency, as it is excreted unchanged mainly through the kidneys [18]. The available literature indicates that acamprosate modulates glutamatergic transmission by affecting N-methyl-D-aspartic acid (NMDA) and metabotropic glutamate-5 receptors, which is the probable mechanism of action of acamprosate. In addition, this compound may also indirectly modulate GABA receptor transmission [19]. Acamprosate, acting by modulating receptors, reduces alcohol craving and unpleasant withdrawal symptoms [20]. A meta-analysis showed that acamprosate improved abstinence rates in alcohol-dependent women and men compared with a placebo, as well as physician cooperation, and that its use was associated with significantly higher treatment completion rates [21]. A Cochrane review of 6915 patients found that acamprosate reduced alcohol consumption compared with a placebo [22]. Acamprosate is used primarily to achieve and maintain complete abstinence rather than as a means to reduce or prevent relapse in regular alcohol use. Its use as an adjunct to psychosocial interventions may help improve abstinence maintenance outcomes in addicted individuals [23].

### 3.2. Nalmefene

Nalmefene is the first drug approved by the EMA for the treatment of alcohol dependence syndrome in adults [24]. The mechanism of action of nalmefene is based on the fact that it is a selective modulator of opioid receptors; it acts mainly as an antagonist of µ- and δ-receptors and as a partial agonist of κ-receptors [25]. The effect on µ-receptors prevents the sensation of pleasure associated with alcohol consumption, while the effect on κ-receptors reduces the dysphoria associated with alcohol withdrawal [26]. The most commonly reported adverse effects of nalmefene include dizziness, nausea, fatigue, headache, pharyngitis, sleep disturbances and insomnia, vomiting, hyperhidrosis, increased appetite, and tachycardia [27]. In clinical trials with addicted patients, nalmefene has been shown to be a well-tolerated, effective drug for the treatment of alcohol dependence syndrome under clinical conditions [28]. In studies with addicted patients suffering from psychiatric disorders, nalmefene, when used as needed, has been shown to reduce alcohol craving in both study groups [29]. In a multicenter randomized trial in Japan, it was shown that alcohol-dependent patients with positive prognostic factors, i.e., no family history of alcoholism and no nicotine dependence, may benefit from the use of this drug [30].

### 3.3. Disulfiram

The probable mechanism of action of disulfiram is its inhibition of the enzyme aldehyde dehydrogenase, resulting in an increase in the plasma acetaldehyde concentration. An increased concentrations of plasma acetaldehyde produces unpleasant sensations after alcohol consumption, i.e., tachycardia, shortness of breath, tachypnea, sensation of heat, anxiety, panic, headache, nausea, and vomiting [31]. In the past, disulfiram was the first FDA-approved drug for the treatment of alcohol dependence syndrome [32]. The available meta-analysis showed that disulfiram remains a more effective drug in open-label studies compared with other pharmacological methods (naltrexone and acamprosate) that can support abstinence. However, double-blind studies have not confirmed its greater efficacy compared with control groups [33]. The major limitation to taking disulfiram is the patient cooperation, so the greatest efficacy is observed with supervised use [34]. In turn, the Agency for Quality Research and Healthcare has found insufficient evidence to support the efficacy of disulfiram [35]. This drug can be used in oral form or in the form of a subcutaneous implant. The study conducted showed that patients who abused alcohol during treatment with disulfiram drank less than in the period before the start of treatment, while the dose of the substance used did not correspond to differences in the amount of alcohol consumed [36]. Studies have shown that the effect of disulfiram is based on its deterrent effect and that the best results are obtained when patients receive psychoeducational training while taking the drug, family support, and therapy monitoring [37]. On the other hand, the effectiveness of disulfiram in clinical trials remains ambiguous, as the results of studies on reducing alcohol craving remain inconsistent in part, because patients have difficulty complying with the substance’s adherence [38].

### 3.4. Naltrexone

Naltrexone is a competitive opioid antagonist with a particular affinity for µ-receptors, which is why the reduction of alcohol consumption with the concomitant use of this substance is associated with the suppression of the reward system and the reduction of pleasure experienced after the consumption of alcohol products [39]. Since 1984, naltrexone has been approved by the FDA in medication-assisted therapy for the treatment of alcoholism, including in extended-release form, and since 1994, it has also been used for opioid dependence [40]. The most commonly observed side effects of naltrexone at doses of 50–100 mg include gastroenterological disturbances, joint and muscle pain, sleep disturbances, headaches, and anxiety [41]. In a meta-analysis, naltrexone was found to be more effective than acamprosate in reducing heavy drinking and alcohol craving, especially if patients underwent detoxification and maintained a sufficiently long abstinence period before starting pharmacological treatment [42]. A systematic review examining the efficacy of oral naltrexone and extended-release naltrexone in alcohol-dependent HIV-positive patients showed that drug treatment reduced alcohol consumption and improved viral suppression without significant side effects [43]. In a study of 32 alcohol-dependent patients, extended-release naltrexone was found to be associated with prolonged abstinence due to improved patient compliance and therefore could be considered as first-line therapy for people struggling with the problem of alcohol abuse [44].

### 3.5. Baclofen

Baclofen is a GABA agonist that selectively binds to presynaptic GABA-B receptors, causing hyperpolarization of motor horn cells and a subsequent reduction in the hyperactivity of muscle stretch, clonus, and skin contraction reflexes. This makes baclofen one of the standard drugs for the treatment of muscle spasticity [45]. Preliminary clinical studies suggest that baclofen can suppress withdrawal symptoms in alcohol-dependent patients and effectively maintain alcohol abstinence [46]. This is an interesting substance, because in 2018, it was approved in France for the treatment of alcohol dependence, although the currently available studies do not show differences between baclofen and a placebo, although even the aforementioned studies note that the results of some studies are promising [47,48]. However, there are concerns about the safety of baclofen, which at high doses can cause serious central nervous system side effects such as mania, sleep apnea, or seizures [49]. In view of this, further research on the use of baclofen and to determine its efficacy and safety in the treatment of alcohol dependence seems crucial, and physicians who decide to use this type of therapy should carefully read the available scientific reports.

### 3.6. Topiramate

Topiramate is an antiepileptic drug that is effective in both partial and generalized seizures. Topiramate has also been shown to be significantly effective in preventing migraines, with a marked decrease in the frequency of migraine headaches. In addition, there are reports on the efficacy of topiramate in the treatment of obesity [50]. Topiramate is considered as a drug with a satisfactory safety profile, and patients describe slowing of thinking or mild impairment of cognitive function as the main side effect [51]. How topiramate affects the brain of addicts is not yet clear, but it is known that topiramate is another drug that affects GABAergic transmission, and recent studies on its use in the treatment of alcohol dependence suggest that topiramate has a strong consumption-inhibitory effect, although further studies and follow-up meta-analyzes are needed to confirm this [52]. Considering that one of the side effects of topiramate is weight loss, which is also used as a target for providing the drug, further clinical trials seem warranted, especially in people who have concomitant alcohol dependence and obesity.

### 3.7. Vareniclin

Attempts are being made to use varenicline, a drug used to treat nicotine addiction, to treat alcohol dependence as well. Varenicline is a nicotinic acetylcholine receptor agonist whose neural response is similar to nicotine’s effect, thereby reducing cravings for nicotine. Alcohol and nicotine addiction often co-occur, and there is evidence that neuronal nicotinic acetylcholine receptors play a biochemical role in both alcohol and nicotine addiction. Research findings suggest that varenicline can selectively reduce alcohol consumption safely and with few side effects [53]. The unique pharmacological properties and the main purpose of varenicline mean that further clinical trials should be conducted, particularly in the area of treating the co-occurrence of alcohol and nicotine addiction, if not only because studies suggest that varenicline reduced the amount of alcohol consumed compared to a placebo, increasing the chance of maintaining abstinence [54]. However, clinical trials remain limited, so scientific evidence for the efficacy of this therapy is limited.

### 3.8. Gabapentin

Gabapentin is an oral anticonvulsant, an amino acid developed as a structural analog of GABA and a GABA-ergic modulator of the calcium channel widely used in pain management. Clinical trials with gabapentin show a reduction in alcohol consumption and craving and a reduction in alcohol-related sleep and affect disturbances in the months following alcohol withdrawal, suggesting therapeutic potential in alcohol use disorders [55]. Gabapentin avoids more days of heavy alcohol consumption (>5 standard drinks per day) than placebo (27% vs. 9%) and may be used as second-line therapy, although there is mixed evidence and concern about the abuse and abuse-related harms of this substance as a stimulant in people struggling with psychoactive substance dependence. Abuse of gabapentin in the general population is approximately 1%, and it is 15% to 22% in patients who abuse opioids; at the same time, the theoretically possible phenomenon of gabapentin abuse in alcohol abusers is unknown [56].

## 4. Possible Future Therapies

### 4.1. Psylocybin

Psilocybin is the main psychoactive substance present in some species of mushrooms found worldwide. Due to its agonism at the 5HT2a receptors, individuals who choose to ingest psilocybin experience a state of altered consciousness, increased introspection, hypnagogic experiences, and perceptual changes such as synesthesia, delusions, and alterations in the sense of time. Psilocybin is considered a safe substance because it is characterized by low toxicity and minimal side effects, and it is not addictive. Therefore, its use is being researched worldwide, including in the treatment of depressive disorders [57]. Some authors suggest that psilocybin may facilitate behavioral changes in people with substance use disorders [58]. These conjectures are supported by the recent 2022 randomized clinical trial demonstrating that psychotherapy with psilocybin leads to a significant reduction in heavy drinking days in people with alcohol problems, outperforming active placebos and psychotherapy [59]. It should be remembered that despite the promising results, there are few such reports, and it is currently still an experimental concept.

### 4.2. 3,4-Methylenedioxymethamphetamine (Mdma, Ecstasy)

3,4-Methylenedioxymethamphetamine, most commonly referred to as MDMA or ecstasy, is a phenylethylamine that releases a large amount of serotonin into synaptic clefts approximately 60 min after oral ingestion, causing a sense of oneness with the world, increased empathy, and heightened sensations to audiovisual stimuli [60]. The effect of this compound is multifaceted, as it not only causes a sudden increase in serotonin, but also acts directly on the α2-adrenergic, serotonin, histamine, β-adrenergic, and dopamine D1 and D2 receptors, and alters the levels of prolactin, oxytocin, cortisol, adrenocorticotropic hormone, and vasopressin [61]. The recent phase III clinical trials confirmed the safety and high efficacy of MDMA in the treatment of post-traumatic stress disorder [62]. In 2017, Ben Sessa proposed the use of MDMA to support the treatment of alcohol dependence syndrome, suggesting that MDMA-assisted psychotherapy (MDMA-AT) may be useful in the treatment of alcohol use disorders due to its ability to enhance the psychotherapeutic process and treat psychological trauma [63]. Initial clinical observations from 2022 suggest that MDMA-AT may also lead to subclinical improvement in alcohol use in severe PTSD without increasing the risk of illicit drug use. Thus, there is preliminary evidence to support the development of MDMA-AT as an integrated treatment for PTSD, alcohol comorbidity, and substance use disorders [64]. Further research in this direction is promising.

## 5. Discussion

Psychiatrists who treat people who are addicted to alcohol believe that psychotherapy gives the best results. However, sometimes it is insufficient, so it should be remembered that there are pharmacological methods to support this type of therapy that can be beneficial for people with alcohol problems. To date, however, there is no drug that ensures significant efficacy and could be called a breakthrough in the pharmacotherapy of alcohol dependence. Studies in 2001 showed that both naltrexone at a dose of 0.1 mg/kg body weight and acamprosate at a dose of 200 mg/kg body weight significantly reduced alcohol consumption, while the combination of these two drugs did not provide a greater benefit, leading to the recommendation that the combined use of acamprosate and naltrexone did not provide better results than monotherapy with these agents [65]. In another meta-analysis comparing the efficacy of alcoholism treatment with nalmefene, naltrexone, acamprosate, baclofen, and topiramate, it was shown that it can be claimed indirectly that topiramate is more effective in the treatment of alcohol dependence syndrome, while its safety profile remains poor, while direct evidence of high efficacy of drinking control by pharmacological treatment with the above-mentioned substances was not provided [66]. It is worth noting that in a study conducted in 2015, it was found that in Europe, only 10% of alcohol abusers receive pharmacotherapy, and when using nalmefene in two randomized, placebo-controlled trials conducted over a 6-month period, it was shown that taking this substance can help control this disease, as the percentage of people who responded to the treatment was higher than in the group of people who received a placebo [67]. The treatment of addictions, not only those from alcohol, causes many difficulties. In studies conducted on people addicted to nicotine, despite the existence of relatively safe and effective therapies to quit smoking, patients repeatedly return to smoking cigarettes and successfully quit smoking after an average of several attempts (a different number depending on data from different countries). This suggests that, in addition to supportive pharmacological treatment, therapeutic interventions seem to be indispensable [68,69]. So far, no study has been conducted that would clearly define the advantage of currently available registered therapies over unregistered ones. More research should be conducted to introduce new pharmacological methods to assist in the treatment of alcohol dependence syndrome and early psychoeducational interventions [70].

## 6. Conclusions

There are effective pharmacological interventions that can help those who need help maintaining alcohol abstinence, although their effectiveness is often limited and there is even limited evidence of the safety or efficacy of some substances, so further clinical research is needed in this area as it is an important social problem from a public health perspective.

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
