# Peer review of "Medications for the Treatment of Alcohol Dependence—Current State of Knowledge and Future Perspectives from a Public Health Perspective"

_ijerph, 2023, doi:10.3390/ijerph20031870_

Round 1

Reviewer 1 Report

This review provides a necessary and convenient summary of the literature on pharmacological approaches to treating alcohol dependence. The review was generally comprehensive and clearly written. I've offered suggestions for strengthening the paper.

1. Introduction:

The introduction clearly defines AUD and adequately addresses the costs to society attributable to alcohol consumption. However, I think the section on health impacts could be strengthened. First, you cite a single meta-analysis on the relationship between drinking and risks of heart failure and leave readers with the impression that low-level consumption (i.e., up to one drink per day) may be protective. While that protective effect may be accurate within the narrow confines of risk for heart failure, there's considerable evidence that even low-level consumption is harmful to other aspects of cardiovascular health. I strongly suggest you cite this review, which discusses the entire cardiovascular system:

Piano, MP. Alcohol’s effects on the cardiovascular system. Alcohol Res 2017;38(2):e1-e24. [url: https://arcr.niaaa.nih.gov/volume/38/2/alcohols-effects-cardiovascular-system]

I suggest also that you divide the introduction into two paragraphs, the first of which (lines 27-49 (through "...systematic review.[8]")) outlines the costs and risks, and the second of which (line 49 (from "Despite the fact...") through line 64) discusses the issues involved in finding a pharmacological solution and outlines the purpose of your review.

2. Methods:

Unless I'm missing something, it appears that you severely restricted your literature search by including in your search terms only the specific drugs mentioned in the introduction (i.e., acamprosate, disulfiram, naltrexone, nalmefene, baclofen, topiramate, varenicline, and gabapentin) as well as psylocybin and MDMA/ecstasy, and "adding" the conditions treated (i.e., alcohol dependence, alcohol abuse, alcoholism). This strategy appears to reverse the usual approach to systematic reviews in which authors do a broad search (e.g., "psychopharmacological treatment" AND ("alcohol abuse" OR "alcohol dependence" OR "alcoholism"), which might yield information on treatments not enumerated in your terms. Similarly, most reviews incorporate more than one database in the search (e.g., MEDLINE, Cochrane, International Pharmaceutical Abstracts, etc.) to ensure casting the widest possible net. I'm not arguing that your approach was inadequate to meet your specific objective. I do think it may be a limitation that needs to be acknowledged because it's possible that viable therapies were not uncovered.

Rather than say that you searched for information on "each substance" (line 70), please list the specific search terms you used, including psylocybin and ecstasy. It would be best to present a table/figure that shows the entire search strategy so that your work could be replicated.

Inclusion of information from reliable web sources is a strength.

3. Results:

Please re-label section 3 as "Results" rather than "Therapy registered for use." Your intention in this section is to present your findings. You may begin the section with a statement that "We found four therapies (acamprosate, disulfiram, naltrexone, nalmefene) registered for use in treating alcohol dependence, four therapies (baclofen, topiramate, varenicline, and gabapentin) not yet registered, and two (psylocybin and MDMA/Ecstasy) possible future therapies." You should then combine current sections 3 and 4 into one (Results) and list/discuss each substance as you do now.

I'm concerned that you may have presented the information about some substances in a way that contradicts your stated intention "to review existing and potentially future pharmaceuticals for the treatment of alcohol dependence in a way that is understandable even for people unfamiliar with reading statistical data." I'm guessing that some readers may be unfamiliar with the details of pharmacokinetics and pharmacological actions of neurotransmitters as well.

a. Acamprosate - Discussing the possible mechanisms of action of each substance is important, but how many readers would immediately understand these statements? "The available literature indicates that acamprosate modulates glutamatergic transmission by affecting N-methyl-D-aspartic acid (NMDA) and metabotropic glutamate-5 receptors as the probable mechanism of action of acamprosate. In addition, this compound may also indirectly modulate GABA receptor transmission [15]." The most important effect of acamprosate is its apparent ability to decrease cravings for alcohol after even brief periods of abstinence (e.g., overnight) by reducing uncomfortable withdrawal symptoms. So the net effect of changing the excitatory and inhibitory processes associated with NMDA, glutamate and GABA is to reduce withdrawal-related cravings. That's what readers need to understand.

b. Nalmefene: You explained well the practical effects of altering the actions of mu, sigma and k receptors.

c. Disulfuram: You state that "The probable mechanism of action of disulfiram is its inhibition of the enzyme aldehyde dehydrogenase, resulting in an increase in plasma acetaldehyde concentration associated with unpleasant sensations after alcohol consumption, i.e., tachycardia, shortness of breath, tachypnea, sensation of heat, anxiety, panic, headache, and nausea and vomiting [26]." I suggest breaking the sentence into two; the first would end with "...acetaldehyde concentrations." The second sentence would explain that increased concentrations of plasma acetaldehyde produce (rather than are 'associated with') unpleasant sensations when alcohol is consumed. I think that makes the connection clearer.

I'm not sure you adequately conveyed the mixed signals on whether disulfuram is effective. Please stress that the evidence is contradictory.

 d. The findings on Naltrexone, Baclofen, Topiramate, Varenicline, Gabapentin, psylocybin and MDMA/Ecstasy are presented clearly, with a couple of exceptions mentioned below in my general comments about wording.

4. Discussion and Conclusion:

You did a good job of stressing that psychotherapeutic approaches appear to be most effective in treating alcohol dependence, and that medications are at best adjunctive treatments that could enhance the effects of psychotherapies. I think you could strengthen this section in two ways:

a. Point out that, so far, we've been less successful at treating addictive behaviors across the board than other forms of psychological/biochemical illnesses. Provide the example of nicotine replacement therapy, which is very safe and only moderately effective. Perhaps draw a parallel between quitting smoking, which often requires multiple attempts (see Chaiton et al., 2016: https://pubmed.ncbi.nlm.nih.gov/27288378/) and quitting drinking. I think it's imperative that readers realize that overcoming alcohol dependence is, in context, similar to overcoming any other addictive behavior, so the relative lack of success in finding a panacea pharmacological approach is not surprising.

b. Given the relative strengths and weaknesses of the registered and unregistered treatments, do you have any sense of how they would rank in the balance of effectiveness and safety, based on current evidence? I realize there's no silver bullet, but perhaps you could offer an evidence-based assessment of relative worth. If not, please say so.

Here are some errors and issues with wording and other 'housekeeping':

Lines 74-75: "The introduction should briefly place the study in a broad context and highlight why it is important." This sentence appears to be an author instruction in the template for manuscripts submitted to IJERPH; it should be omitted.

Lines 143-146: "In meta-analysis, naltrexone was found to be more effective in reducing heavy drinking and alcohol craving, especially if patients underwent detoxification and maintained a sufficiently long abstinence period before starting pharmacological treatment [36]." You've made a comparative statement (i.e., 'more effective') without a comparator. Please add a comparator (i.e., more effective than what?).

Lines 186-192: "Because alcohol and nicotine addiction often co- occur and there is evidence that neuronal nicotinic acetylcholine receptors play a biochemical role in both alcohol and nicotine addiction, attempts are being made to use varenicline, a drug used to treat nicotine addiction that is a nicotinic acetylcholine receptor agonist, to treat alcohol addiction, research findings suggest that the selectivity of this compound in reducing ethanol consumption, combined with a comprehensive safety profile and low side effects in humans, suggests that varenicline may be a drug that can support the treatment of alcohol dependence [47]." Wow! First, this is a massive run-on sentence. Second, I think it needs to be re-worded to be clearer. E.g., "Attempts are being made to use varenicline, a drug used to treat nicotine addiction, to treat alcohol dependence as well. Varenicline is a nicotinic acetylcholine receptor agonist whose neural response is similar to nicotine's effect, thereby reducing cravings for nicotine. Alcohol and nicotine addiction often co- occur and there is evidence that neuronal nicotinic acetylcholine receptors play a biochemical role in both alcohol and nicotine addiction. Research findings suggest that varenicline can selectively reduce alcohol consumption safely and with few side effects [47]."

Lines 227-230: The subsections on psylocybin and MDMA need to be separated and the subsection on MDMA needs to be labeled appropriately (i.e., MDMA/Ecstasy (3,4-methylenedioxymethamphetamine)).

Line 270: The numbering for Conclusions is incorrect. The discussion is #6, which would make the conclusion #7. However, if you follow my suggestion of combining current sections 3 and 4 into a single results section, then current section 5 (possible alternatives) would become #4 and the discussion and conclusion would become #s 5 and 6, respectively.

Author Response

Dear Reviewer, 
Please, see the attachment. 
Best regards, 
Aurthors

Reviewer 2 Report

This narrative review provides useful information on the current knowledge on the use of drugs to treat Alcohol Use Disorder.  The authors have collected information on the human studies carried out in this field and they present  it in an accessible manner to both experts and non-experts which was one of their aims.

I would like to suggest a few points to add to the manuscript:

My major comment is that this review focuses entirely on human research while a lot of work has been carried out in animal models to determine the the effect and mechanism of action of most of the drugs referred to in this  review. While a detailed description of animal model work may be beyond the scope of this review, it would beneficial to at least inform the readers that this work is progressing and a few  relevant references could be added.

There are few other minor modifications to be made :

1) Line 45  As the WHO recommendation seem to  contradicts reference 5 it would be useful to comment to what level of drinking is the WHO referring to with regards to heart disease

2) Line 51 .  AA is only one of many different group support strategies and they do not all follow the 12 Steps. There is substantial literature for example on Peer Support:  Peer-Delivered Recovery Support Services for Addictions in the United States: A Systematic Review https://pubmed.ncbi.nlm.nih.gov/26882891/     This point is partially addressed with reference (10) (line 56), but could be expanded .

3) line 205-211. Something is missing in this very long paragraph. It is not clear 

4) Line 228 The formatting of the subtitle is wrong

Author Response

Dear Reviewer, 
Please see the attachment. 
Best regards, 
Authors

Reviewer 3 Report

Our current model is not succeeding -and needs revision . This paper should
be published, but caveats on the need for evidence-based assessment and
treatment classes according to pharmaceutical effect and value of the drug
is needed. I've enclosed sample ideas.

I view the alcoholism is a symptom of neuropsychiatric and balances, and
patients that are alcoholic frequently switch to nicotine and many other
addictions, including poly addiction. Once patients develop a reward
deficiency  like alcoholism or they tend to need medication and their drive
or urge to self medicate overrides reason and small health or
pharmaceutical interventions like those reviewed. So I like the paper
because it reviews medication, but it adds to another pile of growing
literature, in which the lack of assessment, the lack of sharing of data
and the lack of classifying medication's by their pharmaceutical effect is
absent -

For example, it's known that GABA agents with stabilize. The brain can be
more helpful and patience or diagnose borderline that are on the
Myers-Briggs perceivers, and in general extroverted and volatile .
individuals that are at detached dysthymic depressed often use the
stimulant class of the catecholamine class of medicines can be more helpful
and obviously serotonin agents in individuals that are compulsive,
insomnia, depression, methods that standardize the work up and the
treatments are needed and this at the minimum should be the caveat to this
paper

A person can climb out of many dependency  holes, but the hole of addiction
once  deeply fallen into requires medical " life " support. . Prediction of
that event requires more sophisticated evidence precision based assessment
 with  multiple brain, domains, genetic neurophysiological, cognitive
attention, type temperament, psychiatric tendencies, personality, etc..
 urges, and urges alone  are hard to detect by self report because the
patient has no insight  , and reliable, reproducible data was not in this
paper ( in my opinion )   .
patients  and people have multiple addictions and urges  throughout their
lifetime and switch from one addiction or dependency to another . A
significant brain health check  with evidence based treatment decisions is
just essential to the addiction. Tendencies and urges in the uniform
testing is the only manner by which adequate algorithms can be developed.

Proposing a "Brain Health Checkup (BHC)" as a Global Potential "Standard of
Care" to Overcome Reward Dysregulation in Primary Care Medicine: Coupling
Genetic Risk Testing and Induction of "Dopamine Homeostasis" 2022 Apr
30;19(9):5480.

Assessment  and guide ideas for long term Neuropsychiatric addiction
alcoholism and mental health problems  require standardization of tests and
evidence base to approaches or precision medicine based

Author Response

Dear Reviewer, 
Please see the attachment
Best regards, 
Authors

Round 2

Reviewer 1 Report

Thank you for addressing all of my concerns. Thanks also for explaining why you chose to do a narrative rather than a systematic review. I think you've achieved your objectives.